# Coherent manipulation of valley states at multiple charge configurations of a silicon quantum dot device

Joshua S. Schoenfield[1], Blake M. Freeman[1] & HongWen Jiang[1]

Qubits based on silicon quantum dots are emerging as leading candidates for the solid-state implementation of quantum information processing. In silicon, valley states represent a degree of freedom in addition to spin and charge. Characterizing and controlling valley states is critical for the encoding and read-out of electrons-in-silicon-based qubits. Here, we report the coherent manipulation of a qubit, which is based on the two valley states of an electron confined in a silicon quantum dot. We carry out valley qubit operations at multiple charge configurations of the double quantum dot device. The dependence of coherent oscillations on pulse excitation level and duration allows us to map out the energy dispersion as a function of detuning as well as the phase coherence time of the valley qubit. The coherent manipulation also provides a method of measuring valley splittings that are too small to probe with conventional methods.

[1] Department of Physics and Astronomy, University of California Los Angeles, Los Angeles, California 90095, USA. Correspondence and requests for materials should be addressed to H.J. (email: jiangh@physics.ucla.edu)

Gate-defined quantum dots (QDs) in silicon hetero-structures are emerging as a leading candidate for the physical implementation of quantum computing in the solid state, mainly due to the long coherence times of the individual spins in silicon and the potentials for scaling and integration with mainstream classical electronics[1]. During the last several years, coherent control of spin-based qubits in Si QDs has been successfully demonstrated by a number of groups[2-5].

However, the characterization and control of the valley degree of freedom in silicon nanostructures presents a major challenge. Electrons in bulk silicon have six degenerate states, known as valley states, which, when confined to two dimensions, separate into four high-energy states with higher effective mass and two lower energy states. Only the two low-energy valley states are energetically relevant to charge/spin states in QDs. They are nondegenerate with a splitting energy that depends on the microscopic details of the interface[6-9]. The valley states often affect both the encoding and the read-out of qubits[1]. Thus, characterization and control of valley states is important in order to develop a scalable quantum-computing architecture using silicon-based QDs.

Here, we report the coherent manipulation of a qubit based on the two valley states of an electron confined in a silicon quantum dot. Coherent evolution between the states that have a relatively small energy splitting of 20 μeV is excited by a fast electrical pulse, and the results are projected as the occupations of two different charge states for read-out by a nearby charge-sensing channel. We carry out the valley qubit operations at multiple charge configurations of the double quantum dot device. The dependence of coherent oscillations on pulse excitation level and duration allows us to map out the energy dispersion as a function of detuning as well as the phase coherence time of the valley qubit. The energy structure of the valley qubit is similar to spin–charge hybrid qubits[5, 10, 11], and it shares a desirable resistance to charge noise. The experiment shows that the valley states being manipulated are good quantum numbers. The coherent manipulation presented here also provides a method of measuring valley splittings that are too small to probe with conventional methods of magnetospectroscopy.

## Results

**Device description**. Experiments were performed in a device lithographically identical to the device shown in Fig. 1a. After accumulating a 2-dimensional electron gas (2DEG) with a global top gate, quantum dots, where individual electrons can be trapped, are formed by applying confining voltages (labeled $V_L$, $V_R$, etc.) to the side gates. A charge-sensing quantum point contact (QPC), runs adjacent to the dots with a current, $I_{QPC}$. This current is sensitive to the electron occupation of the dots, and abrupt spikes or dips in the transconductance, $G_\varepsilon \equiv \frac{dI_{QPC}}{dV_\varepsilon}$, represent transitions between charge configurations as an electron is either loaded into, unloaded from, or transferred between the dots in response to a small change in $V_\varepsilon \equiv V_L - V_R$. The device was tuned into a triple dot regime. A charge stability diagram (where $G_\varepsilon$ is mapped out as a function of $V_L$ and $V_R$) can be found in Fig. 1b. The regions between charge transitions are labeled by a three-number tuple with the number of electrons in the left, middle, and right dots, ($N_L$, $N_M$, and $N_R$, respectively).

To operate as a qubit, the device is tuned close to a so-called quadruple point, where there is a charge transition from the (0,1,1) configuration to the (1,0,1) configuration. It is important to point out that although there are the three available quantum dots, the states of only a single electron and its transition between the left and middle dots are used to produce qubit behavior. For this reason, and to emphasize the motion of the single electron, in the detailed stability diagram in Fig. 1c and subsequently in the body of the text, we relabel these charge configurations as (0,1) and (1,0), respectively, where the tuple, ($N_L$, $N_M$), represents the left and middle dot electron occupation numbers. Furthermore, we label the electron states associated with the charge configurations (0,1) and (1,0) as $|M\rangle$ and $|L\rangle$, respectively, with subscripts added as needed to refer to specific valley states within these charge configurations.

For qubit operation, the primary control parameter is the middle-left dot detuning, $\varepsilon$, which is a measure of potential energy asymmetry between the two dots induced by the confining gates. Adjusting $\varepsilon$ allows for the electron to transfer between the (0,1) and (1,0) states. As defined in this paper, at positive detuning, the (1,0) configuration is energetically favored and at negative detuning the (0,1) configuration is favored. When detuning is zero, the two configurations are equally favored. This case corresponds to the detuning line in Fig. 1c, represented by a dotted line at the boundary between the (1,0) and (0,1) regions of the stability diagram. The solid arrow represents the directions of increasing and decreasing $\varepsilon$.

If no valley splitting is present, or if its value is too large to be relevant to the dynamics, the energy spectrum of the system as a function of detuning would be that of a standard charge qubit as shown in Fig. 2a. In contrast, Fig. 2b shows the spectrum in the presence of a small valley splitting in the left dot, as in the case in our experiments.

When strongly detuned into the (1,0) charge configuration, the ground and the first excited valley states, $|L_{v_1}\rangle$ and $|L_{v_2}\rangle$, respectively, are separated by the valley splitting $\delta$. The qubit subspace will be the Hilbert space spanned by these two valley states. A model Hamiltonian describing this system, on the basis of $\{|M\rangle, |L_{v_1}\rangle, \text{and } |L_{v_2}\rangle\}$ is given by

$$H = \begin{pmatrix} \frac{\varepsilon}{2} & \Delta_1 & \Delta_2 \\ \Delta_1 & -\frac{\varepsilon}{2} & 0 \\ \Delta_2 & 0 & -\frac{\varepsilon}{2} + \delta \end{pmatrix}, \tag{1}$$

where $\Delta_1(\Delta_2)$ is the interdot tunnel couplings between the middle dot and the left ground (excited) state.

**Interference and coherent oscillations between valley states**. To perform the experiment, positive voltage square pulses were applied to $V_L$ in Fig. 1c in order to induce an interdot electron transition. Applying such a pulse results in the clear interference fringes present in the stability diagram in Fig. 1c. Applying a pulse to only a single gate may temporarily bring the system to a part of voltage space outside of either the (0,1) or (1,0) region. However, because the tunneling rates to the electron reservoirs are slow relative to the pulse durations, which are a few nanoseconds at the longest, the electrons are unlikely to be either loaded or unloaded from the dots during the pulse. Therefore, in Fig. 1c, the detuning line is extended into the regions (0,0) and (1,1) to represent that these states are generally invisible to the qubit during its operation time, but that this line continues to represent the boundary between voltage regions, where (0,1) and (1,0) are preferred relative to one another. The effect of applying a simultaneous negative voltage pulse to $V_R$ is discussed in Supplementary Note 2.

The generated pulses were nominally square, starting at an initial detuning, $\varepsilon_0$, and nominally rising to a detuning of $\varepsilon_p$ for a pulse time $t_p$. But, crucially, in practice they had a nonzero rise

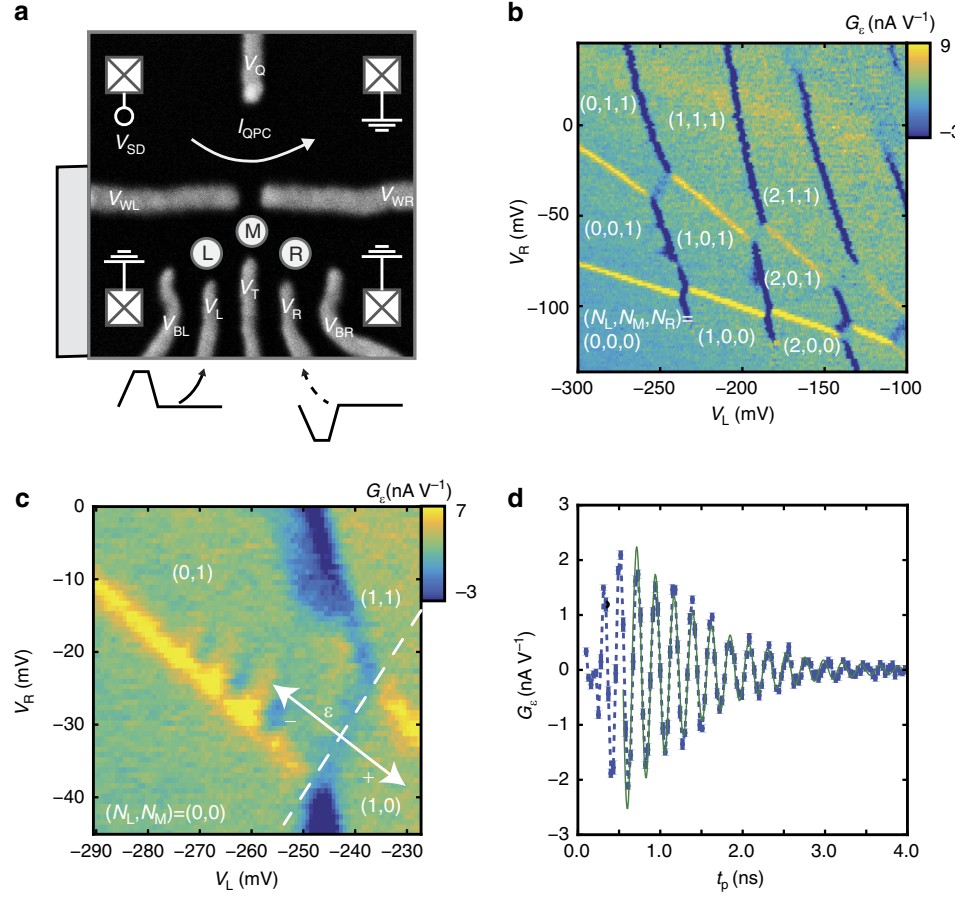

**Fig. 1** Observation of coherent effects. **a** A scanning electron micrograph of the quantum-dot-forming region in a lithographically identical device: The three circles represent the estimated location of three dots. The squares represent ohmic contacts. The *grey scale bar* pictured at left represents 500 nm. **b** The *charge stability diagram*, which maps out the ground state electron occupancy of the three dots as a function of side gate voltages: **c** A close-up of interference fringes at the $(0,1,1) \leftrightarrow (1,0,1)$ charge transition: A 500 ps square pulse with an amplitude of +30 mV is placed on $V_{\mathrm{L}}$. The 0 detuning line is shown as a dotted line and the detuning axis is shown as a solid arrow. **d** Time domain oscillations: Fixing $V_{\mathrm{L}}$ and $V_{\mathrm{R}}$ at values within the interference region and fixing the pulse height at +24 mV, the nominal time of the square pulse is varied, resulting in an oscillatory average charge occupation. *Error bars* are the one standard deviation range for that data point as taken over 10 averages. Also shown is a fitted decaying sinusoid used to extract a frequency $(4.41 \pm 0.01$ GHz here) and decay time $(0.90 \pm 0.04$ ns), the latter of which serves as a lower bound on $T_2^*$. A moving average with a window of several periods of the oscillation has been subtracted prior to fitting to counteract pulse duty cycle effects

time, $t_{\mathrm{rise}}$. To a reasonable approximation, the detuning rises at a constant rate, $\nu \equiv \frac{d\varepsilon}{dt} \approx \frac{\varepsilon_p - \varepsilon_0}{t_{\mathrm{rise}}}$. The pulse then reaches a maximum detuning, $\varepsilon_{\max}$, where it remains for a time $t_{\max}$, corresponding to the remainder of $t_{\mathrm{p}}$. (If the rise time is longer than $t_{\mathrm{p}}$, then $t_{\max}$ might be zero. If $t_{\mathrm{p}} > t_{\mathrm{rise}}$, then $\varepsilon_{\mathrm{p}} = \varepsilon_{\max}$.) After $t_{\mathrm{p}}$ has elapsed, the detuning returns to $\varepsilon_0$, again at a rate of roughly $\nu$, where it remains for 30 ns until the next pulse.

The clear interference fringes in Fig. 1c induced by a short pulse of 500 ps are a signature of the coherent evolution between the two valley states in the left QD during the pulse duration (detailed discussion in next section). We believe that the energy splitting in the middle dot is much larger, as no interference can be observed when the pulse is pumped in the opposite direction.

Fixing the DC voltages of the side gates at a point within the interference region, coherent time domain oscillations of $G_\varepsilon$ are observed by varying pulse width. Figure 1d shows the differential measurement QPC current, which relates to the probability of being in state $(0,1)$, $|\mathrm{M}\rangle$, at the end of operation time, as a function of $t_{\mathrm{p}}$, for a given pulse height. Also shown as a solid line is a decaying sinusoid fit used to extract both the oscillation frequency and decay times, the latter of which serves as a lower bound on $T_2^*$.

**Principle of the valley qubit operation and read-out**. The qubit operation is outlined in Fig. 2c–f. To avoid possible confusion, we emphasize here, for this experiment, that the left QD is the operation QD, where qubit operation takes place, and the middle QD is the measurement QD, where the projective read-out is performed. In step 1, the system begins at a negative initial detuning, $\varepsilon_0$, with the electron in the ground state, $|\mathrm{M}\rangle$. Then, so long as the level anticrossing between $|\mathrm{M}\rangle$ and at least one of the left states is highly gapped relative to $\sqrt{\hbar\nu}$, the pulse adiabatically brings the electron from state $|\mathrm{M}\rangle$ to the left dot. As the pulse continues to rise, the system is brought nonadiabatically through a level anticrossing that occurs at detuning $\varepsilon_x$. Near this anticrossing, the lower two eigenstates are both superpositions of $|\mathrm{L}_{\mathrm{v}_1}\rangle$ and $|\mathrm{L}_{\mathrm{v}_2}\rangle$, a feature that is indicated in yellow on the spectra in Figs. 2b, 3b and d. The total effect of these two transitions is a unitary transformation that loads the state $|\mathrm{M}\rangle$ into a state approximately given by $e^{i\Phi(0)}\left(\cos\left(\frac{\theta_{\mathrm{load}}}{2}\right)|\mathrm{L}_{\mathrm{v}_1}\rangle + e^{i\phi_{\mathrm{load}}}\sin\left(\frac{\theta_{\mathrm{load}}}{2}\right)|\mathrm{L}_{\mathrm{v}_2}\rangle\right)$, where $\theta_{\mathrm{load}}$ and $\phi_{\mathrm{load}}$ are angles in the Bloch sphere defined by $|\mathrm{L}_{\mathrm{v}_1}\rangle$ and $|\mathrm{L}_{\mathrm{v}_2}\rangle$, and $\Phi(0)$ is an overall phase that does not affect measurement. This unitary transformation as well as the overall loading fidelity into the left subspace are dependent on $\nu$.

In step 2, once the pulse reaches its maximum detuning, $\varepsilon_{\max}$, the system is far detuned toward the left dot, and assuming that

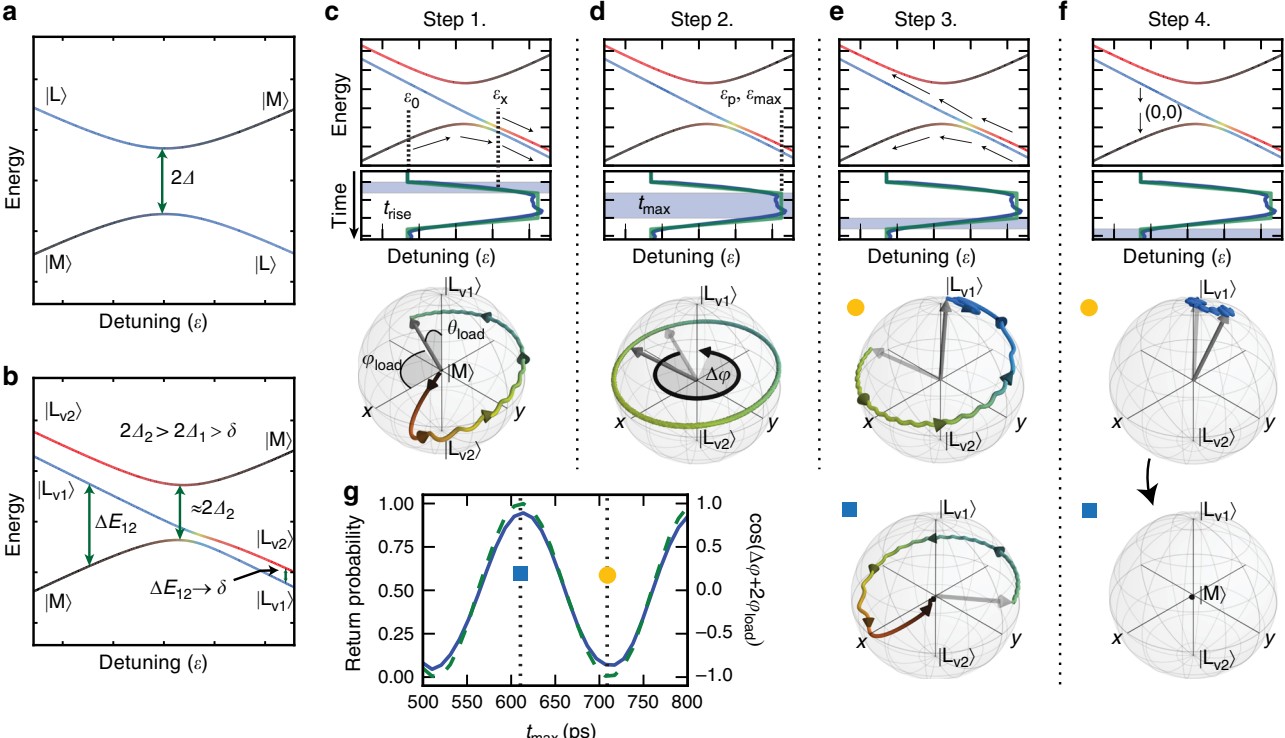

**Fig. 2** Principle of operation and read-out. **a** The two-state spectrum of a standard charge qubit Hamiltonian with tunnel coupling $\Delta$, and without valley states. **b** The three-state spectrum in the presence of a small valley splitting, $\delta$ (Not to scale). The same coloring is used in these spectra as in Fig. 3b and d. **c–f** The steps of qubit operation: The detuning as a function of time is shown as a *solid green line* in relation to the three state spectrum (solid blue line shows a realistic pulse profile). The *blue shaded* region of time is the portion of the pulse that makes up a given step. Also shown is the evolution of the state during each step within the Bloch sphere of the left dot valley states. The paths have been colored in the same manner as the spectra. **c** In step 1, the loading step, the detuning ramps up over a time, $t_{rise}$, from its initial value, $\varepsilon_0$, through a level crossing at $\varepsilon_x$. The final state has Bloch coordinates $\theta_{load}$ and $\phi_{load}$. **d** In step 2, the accumulation step, the detuning stays at $\varepsilon_{max}$ ($\varepsilon_p$ for pulses of sufficient duration) for a time $t_{max}$, during which time it accumulates a dynamical phase $\Delta\phi$. **e** In step 3, the read-out step, the detuning returns to $\varepsilon_0$. Two state evolutions with different $t_{max}$ are shown, one labeled with a yellow circle and the other with a blue square. **f** In Step 4, the measurement/re-initialization step, the system relaxes to |M⟩, aided by temporarily transitioning to charge configuration (0,0). **g** The simulated probability of the electron returning to state |M⟩ at the end of read-out is plotted along with $\cos(\Delta\phi + 2\phi_{load})$ as a function of $t_{max}$, showing that the total accumulated phase is encoded in the return probability. Also marked are the two values of $t_{max}$ for the evolution pathways displayed in **f** and **g**. The system marked with a *yellow circle* has accumulated an additional $\pi$ radians of phase relative to the system marked with a *blue square*

$t_p > t_{rise}$, the detuning level is now constant for a time $t_{max}$, so the state will evolve according to the Schrödinger equation as

$$|\Psi\rangle = e^{i\Phi(t)}\left(a|\psi_1\rangle + e^{-i\frac{\Delta E_{12}(\varepsilon_{max})}{\hbar}\Delta t}b|\psi 2\rangle\right). \quad (2)$$

Given that near $\varepsilon_{max}$, the eigenstates $|\psi_1\rangle$ and $|\psi_2\rangle$ are almost exactly the states $|L_{v_1}\rangle$ and $|L_{v_2}\rangle$, this may be approximated as

$$|\Psi\rangle = e^{i\Phi(t)}\left(a|L_{v_1}\rangle + e^{-i\frac{\Delta E_{12}(\varepsilon_{max})}{\hbar}\Delta t}b|L_{v_2}\rangle\right), \quad (3)$$

where $|\psi_1\rangle$ and $|\psi_2\rangle$ are the ground and first excited state, respectively, $\Delta E_{12}$ is their energy difference, $a$ and $b$ are the initial coefficients when $\varepsilon_{max}$ is first reached, $\Delta t \equiv t - t_{rise}$ is the time spent at $\varepsilon_{max}$, and $\Phi(t)$ is a time-dependent overall phase that does not affect measurement. The axis of control is almost purely along the $z$-axis of the Bloch sphere. Therefore, the state processes around the $z$-axis at a Larmor frequency given by $\Delta E_{12}$. The effect on the qubit state of the system being maximally detuned for a time $t_{max}$ is approximately given by $R_z(\Delta\phi)$, where $R_z(\phi) \equiv e^{i\phi\sigma_z}$ and $\Delta\phi = \frac{\Delta E_{12}}{\hbar}t_{max}$.

In step 3, as the pulse returns to its initial value, $\varepsilon_0$, the system first pulses through the anticrossing at $\varepsilon_x$, experiencing another nonadiabatic transition. The system then experiences another

adiabatic charge transition, where the ground state is projected to |M⟩ and the first excited state is projected to the left states. The net result of this step is that the total phase accumulated, $2\phi_{load} + \Delta\phi$, is encoded in the probability of the electron returning to the middle dot, as shown in Fig. 2g. As these two states correspond to different charge configurations, they are discernible to the QPC, which collapses the system to one of these two measurement states. If the electron is measured to have remained in the left dot, it will remain there until it relaxes back to the ground state. It is worth noting here that the valley relaxation time in a single QD is known to be very long, on the order of microseconds[12–14], several orders of magnitude longer than our pulse repetition time.

Step 4, the measurement/re-initialization phase, is much longer than $t_p$; so its average charge configuration is the primary contributor to the differential transconductance, $G_\varepsilon$. Therefore, $G_\varepsilon$ is sensitive to the probability of the electron finally being in state |M⟩ and ultimately to $\Delta\phi$. Further enhancing the sensitivity of the QPC is the fact that as the charge relaxation rate, $T_1$, is likely longer than the measurement/initialization time of 30 ns, the read-out will be most pronounced in the so-called high visibility region, where the fastest relaxation channel of the electron from the left dot back to the middle dot is by first having the electron tunnel out of the dots ((1,0)→(0,0)) and then a different electron tunneling directly back into the middle dot

$((0,0)\rightarrow(0,1))$. In such a region, the total charge occupancy of the dots temporarily changes, which enhances read-out sensitivity as the QPC is more sensitive to the total number of electrons in the dots than to their positions within the dots. A more detailed discussion of this process is discussed in Supplementary Note 2. After this relaxation process, the qubit has been re-initialized to $|M\rangle$.

We point out that the techniques used in this work to probe valley states bear a strong experimental and formal similarity to the electronic beam-splitter methodology used to probe the (1,1) $S$-(1,1)$T_-$ of a double quantum dot[15].

**Detuning-dependent coherent oscillations**. Figure 3a shows the dependence of the coherent time domain oscillations on $\varepsilon_p$. In Fig. 3b, the energy spectrum is plotted over a detuning range that matches the one explored in Fig. 3a. Fig. 3c shows a simulated lock-in signal produced by time-evolving the Schrödinger equation under the influence of a square pulse with a finite rise time as a function of $\varepsilon_p$ and $t_p$.

The fixed parameters in the Hamiltonian, $\Delta_1$, $\Delta_2$, and $\delta$, are determined by matching the observed frequency of time domain oscillations to the numerically calculated value of $\Delta E_{12}$ as a function of $\varepsilon$ using a weighted least squares fit. In Fig. 3d, the observed frequency is plotted vs. detuning along with the calculated value of $\Delta E_{12}$.

An example of the decaying sinusoidal curve fit used to extract the observed frequency is presented in Fig. 1d as the solid

green line. The fit was restricted to data where $t_p$ was greater than 500 ps, to allow the pulse to significantly exceed its rise time, which is estimated to be about 100–200 ps. By making this restriction, a change in the total pulse time, $\Delta t_p$, leads directly to a change in the time spent at the maximum detuning, $\Delta t_{max}$. The additional pulse time should, therefore, contribute directly to additional phase accumulation at detuning $\varepsilon_p$. The extracted frequency can then be used to map out the detuning dependence of $\Delta E_{12}$, as shown in Fig. 3d.

The extracted Hamiltonian parameters in general depend on the lever arm, $\alpha$, which, based on magnetospectroscopy of a lithographically identical device in a similar configuration, we estimate to be 3%. The only extracted parameter that is nearly independent of $\alpha$ is $\delta$, which is measured reliably to be ~5.6 GHz across a variety of assumed $\alpha$'s. This is because $\delta$ is the value that the observed frequency approaches as the dot becomes highly detuned (see the dotted line in Fig. 3d).

Through simulation, we find that the specific value of $\alpha$ assumed and the corresponding values for $\Delta_1$ and $\Delta_2$ in the Hamiltonian affect the details of evolution in a detuning region ($\varepsilon_p < \varepsilon_x$) where $\left|\frac{d\Delta E_{12}}{d\varepsilon}\right|$ is large. In this region, susceptibility to charge noise is too great to observe coherence, given the noise level of our experimental setup. In the highly detuned region ($\varepsilon_p > \varepsilon_x$) where $\Delta E_{12}$ approaches $\delta$, $\left|\frac{d\Delta E_{12}}{d\varepsilon}\right|$ is small; hence, the qubit is protected against charge-noise-induced decoherence.

The increased coherence times in a wide region of nearly constant dispersion is a feature shared with recently reported spin–charge hybrid qubits, which rely on the small correlation

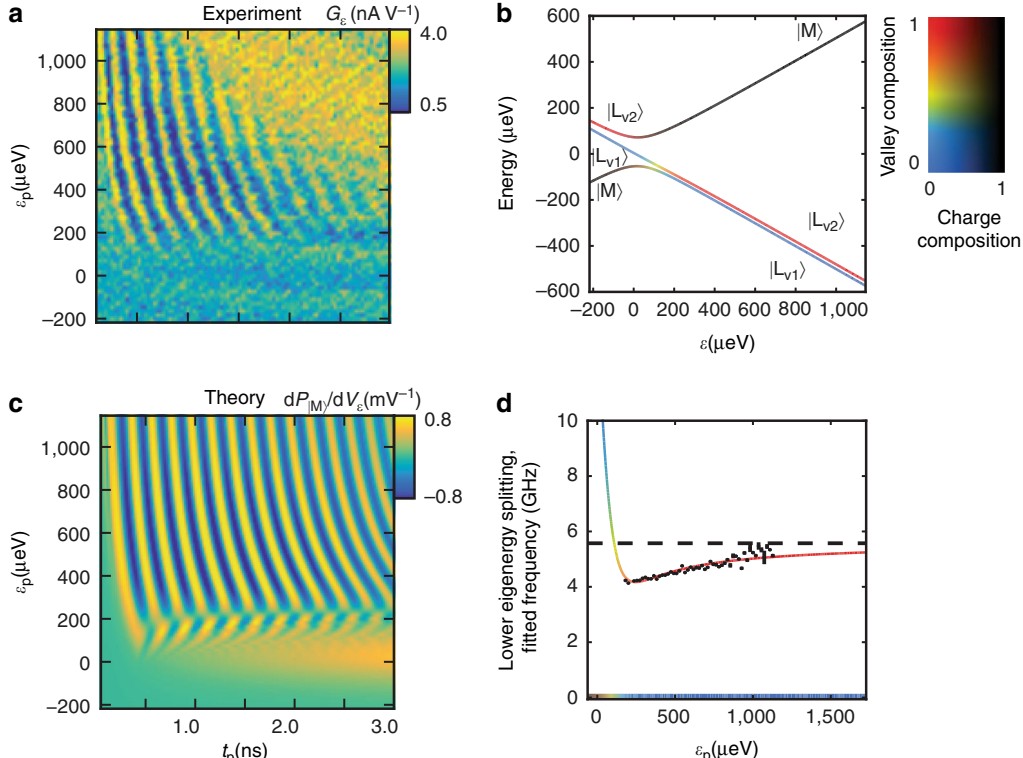

**Fig. 3** Detuning dependence of oscillations. **a** Experimental mapping of the detuning dependence of the time domain oscillations; **b** Energy spectrum derived from fitting to the extracted dispersion: A two-dimensional color bar is used to simultaneously display a state's makeup in the charge degree of freedom, calculated as the probability of a state, $|\psi\rangle$, being in the middle dot, $|\langle\psi|M\rangle|^2$, and the valley degree of freedom, calculated as the probability of being in the first excited valley state conditional upon being in the left subspace, $\frac{|\langle L_{v2}\rangle|^2}{1-|\langle|M\rangle|^2}$. ($\delta = 5.57$ GHz, $\Delta_1 = 6.4$ GHz, $\Delta_2 = 13.6$ GHz) **c** Simulated charge-sensing transconductance signal, obtained by solving the time-dependent Schrödinger equation without dissipation: A square pulse with a finite rise time of 200 ps was assumed. **d** Experimentally extracted frequencies plotted along with the separation between the lower two eigenenergies as a function of detuning: The extracted value of the valley splitting, $\delta$, is plotted as a dotted line. The color of the fitted curve is the color associated with the middle eigenstate at a detuning using the coloring from **b**. Along the bottom axis, the color represents the makeup of the ground state

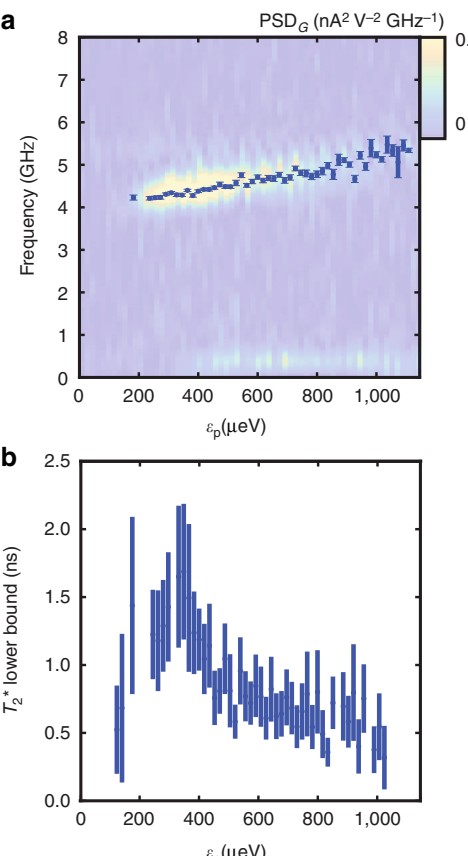

**a**

PSD$_G$ (nA$^2$ V$^{-2}$ GHz$^{-1}$)

**b**

**Fig. 4** Dispersion and decoherence. **a** The dispersion extracted from the transconductance data in Fig. 3a: The points are the frequencies extracted by applying a decaying sinusoidal fit to each cut in $\varepsilon_p$. The background is the magnitude of the power spectral density of those same cuts, calculated using Welsh's method. **b** The extracted values of the decay time from the sinusoidal fit: This value is actually a lower bound on phase decoherence time $T_2^*$. The decay time increases rapidly at first and then begins to decrease. The longest value directly observed is 1.5 ns. The height of the lines in both **a** and **b** indicates the 1.5-$\sigma$ standard errors derived from the numerically estimated covariance matrix

energy of singlet/triplet-like states. However, in hybrid qubits, which tend to be guided by Hamiltonians where $2\Delta_1$, $2\Delta_2 < \delta$, the frequency approaches a minimum close to the detuning line. Instead, in our system, the qubit behavior does not appear until deeply detuned into the left dot, and, even then, it never exhibits the pronounced chevrons as the pulse approaches the detuning line.

**Phase coherence**. From the decaying sinusoid fit of the QPC signal, one can extract the decay time of the oscillations, which is a lower bound on the coherence time $T_2^*$. As shown in Fig. 4b, there is a peak in the observed coherence time at $\varepsilon_p$ of a few hundred μeV. For comparison, the measured dispersion is displayed in Fig. 4a. At smaller detuning levels, the coherence time decreases rapidly. This is likely due to being in a region of rapidly increasing $\left|\frac{d\Delta E_{12}}{d\varepsilon}\right|$, which will enhance the decohering effect of charge noise. Eventually oscillations disappear, likely due to not having experienced the nonadiabatic portion of the pulse.

At higher detuning, the coherence time decreases gradually. This is likely due to the increased tendency of the system to transition to the nuisance (1,1) charge configuration as that state is made more favorable relative to the (1,0) configuration in which the qubit operates. At higher pulse amplitudes, the

potential barrier between the left dot and the electron reservoir is lower, potentially exposing the electron to the effects of the environment.

The maximum observed coherence time is about 1.5 ns. We believe that it is limited by the system tunneling, as the pulse is not applied parallel to the detuning direction. As will be discussed in Supplementary Note 2, there is a evidence that the actual decoherence time could be longer, if the pulse-induced tunneling to the reservoir can be suppressed.

**Discussion**

It is important to point out that coherent oscillations with similar frequencies were observed at two consecutive anticrossings. Such a behavior was present not only at the quadruple point connecting the (0,1,1) and (1,0,1) charge regions but also at that connecting the (1,1,1) and (2,0,1) regions, as can be seen in Supplementary Fig. 1a. The details of this behavior may be found in Supplementary Note 1. The presence of the behavior at multiple consecutive anticrossings provides clues as to the nature of the excited state used in the qubit operation. For instance, the states whose coherent manipulation is observed here are unlikely to be spin-based, as the spectrum of energies associated with spin degrees of freedom is highly dependent on electron number parity.

Furthermore, the possibility of the qubit space being composed of orbital states is precluded by the magnitude of $\delta$. As the charging energy of our QD is about 2.5 meV, a QD radius of 40 nm is expected, which would correspond to the first excited orbital state of 0.4 meV above the ground state. Such splitting is an order of magnitude higher than $\delta$. In addition, the energy splitting observed for the (1,1)–(2,0) transition is very close to that for the (0,1)–(1,0) transition (both are about 6 GHz). If these were orbital excited states, they would change significantly, and tend to decrease, as the electron number is increased. That the splitting remains constant is consistent with the report that valley–orbit coupling is unaffected by the occupation number[16]. Also, the control of oscillations with an energy splitting well below the thermal energy scale suggests that the valley states being manipulated are good quantum numbers. We therefore conclude that the qubit behavior reported here is the result of coherent manipulation of the valley states of silicon.

The magnitude of the energy splitting between valley states in a given silicon-based device is an important value, as it influences whether the device will be suitable for many exchange-based qubit architectures, which in general requires the presence of Pauli spin blockade to perform spin-to-charge conversion. Valley splittings that are too small can disrupt spin blockade by providing an additional low-energy state for an electron to tunnel into with the same spin configuration as that which would otherwise be blockaded. Often the magnitude of the valley splitting in a device is measured using magnetospectroscopy. However, the magnitude of the valley splitting reported here in both charge configurations is below the measurement threshold for such techniques. The coherent measurements performed here could serve as a valuable method of probing the magnitude of certain smaller valley splittings.

Finally, as a host candidate for quantum information processing, qubits operating in a region of detuning invariant energy splitting, such as hybrid qubits and the valley qubit reported here, offer a promising combination of fast operation and resistance to noise.

**Methods**

**Sample preparation**. The sample fabrication is similar to that described in a recent study[17]. Devices were fabricated on a SiGe substrate consisting of a 16 nm silicon well, a 40 nm Si$_{0.7}$Ge$_{0.3}$ spacer, and a 2 nm Si cap. All devices were fabricated by first patterning Ti/Au depletion gates on the substrate. Following this, 100 nm of

Al$_2$O$_3$ was grown using atomic layer deposition to provide an insulating layer. Finally, a 300 nm global top gate was patterned over the device area. Ohmic contacts, depicted as squares in Fig. 1a, are connected to the source and drain of the QPC and are used to bias the charge sensor as well as measure its current. The QPC measures the average charge occupancy of the dots, which is dominated by the occupation configuration during the measurement phase, which comprises upward of 90% of the duty cycle. The QPC transconductance current is measured via a lock-in measurement, where $V_L$ and $V_R$ are simultaneously varied in order to obtain $G_\varepsilon$.

**Simulation**. The evolution of the qubit was examined numerically by time evolving the Schrodinger equation in the density matrix formalism[18]:

$$\frac{d\rho}{dt} = -\frac{i}{\hbar}[H, \rho]. \tag{4}$$

The expected charge state is given by the probability of being in the middle dot: Tr($\rho$|M⟩⟨M|). The simulation explicitly included neither dephasing nor charge relaxation, and could therefore not make a full prediction of the average charge configuration. Instead, the final charge state after 4 ns of operation time is used as a proxy for the average QPC current. As this final time is well into the measurement period for all operation times considered, and, as the charge relaxation time $T_1$ is believed to be relatively slow, this state is taken to be a representative of the pulse's effect on the average charge configuration observed by the QPC and therefore of the average QPC signal.

Simulation of the transconductance signal was achieved by sampling the simulated $\varepsilon_0$ from $\varepsilon_0 + A\sin(x)$, where the spacing in $x$ was even between $-\frac{\pi}{2}$ and $+\frac{\pi}{2}$. Then, the calculated average charge occupancy from these simulations was numerically integrated against $\sin(x)$ to simulate the real part of the QPC signal. The experimentally obtained transconductance, such as that displayed in Figs. 1 and 3, is the real part of a lock-in signal with a modulation frequency of 132 Hz.

**Data availability**. The data that support the findings of this study are available from the corresponding author on reasonable request.

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

## Acknowledgements

This work was supported by U.S. ARO through Grant No. W911NF1410346.

## Author contributions

J.S.S. fabricated the devices and performed the experiment. J.S.S. and B.M.F. conducted the numerical simulation and analyzed the data. H.W.J. supervised the project. All authors contributed to the data interpretation and to the paper writing.

## Additional information

**Competing interests:** The authors declare no competing financial interests.

