## [Peer Review File · Nature Communications]

Reviewers' comments:

Reviewer #1 (Remarks to the Author):

The authors report a very solid and competent set of experiments on a surface-gated silicon structure. They performed good fabrication, and sensitive measurements which they interpret as evidence of the manipulation of valley states in the electrically confined region of the silicon. I think this interpretation is believable, and the data are very clear. They show Rabi oscillations and clear detuning effects. The paper is clearly written. I believe that the results will be of interest to experts in the field, and whilst not providing a viable new qubit technology, due to the small level separation and short coherence time, this work is a useful adjunct to the silicon qubit literature. The issue of valley splitting has been a constant source of discussion in the field, and these results are a step in clarifying that discussion. I recommend that the paper be published as it stands.

Reviewer #2 (Remarks to the Author):

This manuscript presents some interesting data, but has several problems that make it unpublishable in its present form. The biggest problem is that I am not at all convinced that the observed phenomena are attributed correctly to the oscillations of a valley qubit. But there are also deficiencies in the description of the experiment as well as in attribution of previous work that also need to be resolved before the paper could be seriously considered for publication.

First I discuss why I doubt that the oscillations are arising from a valley qubit. The stability diagram shown assigns occupations to the dots, but no data shown make the case for these assignments. The key issue here is that if the left and middle dots were both to have one more electron than assigned, then the occupations of the left and middle dot would be (2,1) and (1,2), and the oscillations would be those of a quantum dot hybrid qubit (see, e.g., Refs. 5 and 10 in the manuscript). The stability diagrams are cut off well before further transitions would be observed, but even if they do not yield additional transitions, it is important to show magnetospectroscopy data to provide some confirmation of the occupation assignments.

The authors state that an interpretation in terms of a quantum dot hybrid qubit is not tenable because similar phenomena are seen if the occupation of the middle dot is increased by one, but the data shown do not convince me of this. Quantum dot hybrid qubit-type oscillations could still occur after the addition of an extra electron to the left dot (because the oscillations are occurring between two states in the middle dot, the electron occupation of which is not changing). It is true that the additional electron on the left could cause a singlet state to form in that dot, but in these systems it is quite possible that the valley splitting for the additional electron is low, in which case it could well be that some spin-related phenomenon such as spin blockade would be observed some of the time and oscillations some of the time, depending on the valley state. This scenario is consistent with the strong signal observed near the polarization line in the presence of the additional electron that is absent when there is no extra electron. I don't think that there is a simple explanation of the marked differences in the data near the polarization line in terms of a single-electron valley qubit. This is exactly the argument given by the authors in reverse: if the oscillations are between valley states, then the data at the two fillings should be similar. Since the data are qualitatively different near the polarization line, that is evidence that the oscillations are not between two valley states.

Another reason why I doubt that the oscillations are between valley states is that the time scales of the experiment are much more typical of those of charge and quantum dot hybrid qubits, while the time scales of valley relaxations have been measured to be much longer. As discussed in the manuscript, the relaxation time is probably of the same order as the pulse repetition time, which is reported in the manuscript as about 30 ns; this value is quite similar to those measured in charge and quantum dot hybrid qubits. The decoherence time of about 1 ns is much closer to that

observed in charge and hybrid qubits, and much less than the time scales for valley relaxations, which have been measured to be of order of tens and even hundreds of microseconds (see next paragraph).

Now I discuss previous work that is relevant that is not mentioned in the manuscript. The first is that the pulse sequence and measurement interpretation is extremely similar to the "electronic beamsplitter" investigated by Petta et al. (Science 327, 669 (2010), plus follow-on work). I agree that the experiments reported in this manuscript are unlikely to be due to the S-T_z oscillations explored in the Petta manuscript, but the experimental method as well as the phenomenology of the two cases have a lot of similarities. The second set of related work is the work that the Vandersypen lab has done measuring valley relaxations: they have showed that valley relaxation in a single dot is many microseconds (there are two papers, one reporting a valley relaxation time of about 50 microseconds (E. Kawakami et al., Nat. Nano. 9, 666 (2014)) and the other reporting a valley relaxation time of hundreds of microseconds (PNAS 113, 11738 (2016))).

To summarize so far, because I think that the interpretation of the experiments is wrong, I do not recommend publication of the current version of the manuscript in any journal.

I now move on to general issues that should be addressed to make the set of measurements more complete and interpretable.

- 1) Magnetospectroscopy measurements should be shown, to provide better evidence of the electron filling factors at which the experiments are performed.
- 2) The relaxation time relevant for readout should be measured, to be able to give some sense of the magnitude of the oscillations compared to one electron moving from one side of the double dot to the other. The authors state that the the relaxation rate may be determined by tunneling out of one dot and into the other. If they have experimental evidence for this claim, it should be shown. If the process is a relaxation process within a single dot, , the dependence of the relaxation rate on detuning will yield insight into the underlying physics (see, e.g., Wang et al., Phys. Rev. Lett. 111, 046801 (2013); Nguyen et al., arXiv:1403.3704). In either case, the dependence of this relaxation rate on detuning should be characterized in order to obtain some information about the strength of the oscillations as the detuning is changed.
- 3) The main text should state that the signal being plotted is the real part of the output of the lockin amplifier (at present this is stated in the supplementary material, but not the main text). The frequency of the modulation should be reported.

A confusing typo:

Supplemental Fig. S1:

(2,0,1) appears twice; I assume that the lower one should be (2,0,0) (modulo my doubts about the assignments made alluded to above).

Reviewer #3 (Remarks to the Author):

This paper claims to observe the coherent manipulation of valley states in a silicon quantum dot by the application of fast gate control pulses to a triple quantum dot system, although in this case only two of the quantum dots appear to contribute to the effect, since the third dot is maintained at a constant occupancy throughout the experiments. The experiment also claims to provide a method to extract the valley splitting with high accuracy.

I believe that there is sufficient evidence in the experimental results to support these claims, but I will return to this point later, and firstly consider the overall potential significance of the paper.

Silicon-based quantum dots are generating significant interest as qubits for future quantum

computing systems because of the long spin coherence times that are available in silicon and their compatibility with industrial silicon CMOS manufacture. In many cases, such spin qubits rely on the process of Pauli spin blockade to discriminate between singlet and triplet two-electron states, which can either be the qubit basis states, or serve as a pathway for readout of single spins.

As the authors correctly note, the Pauli spin blockade process requires a sufficiently large valley splitting, but many silicon quantum dots, in particular those formed at Si/SiGe heterostructures, can have very small valley splittings (much less than 0.1 meV) which make singlet-triplet readout impossible. While this is not a fundamental obstacle to silicon spin based QC (since a number of groups have now demonstrated spin blockade), it is proving to be an issue that is slowing progress in the field, and is therefore of considerable practical interest to the silicon QC community.

I therefore believe that the spin qubit community will find this paper both interesting and useful, and the experiments and analysis here will provide a useful tool to characterise small valley splittings when they occur.

I also believe that this work is indeed novel, as I am not aware of any other experiments which have shown coherent oscillations that can be attributed to valley states.

I now return to the question of whether the observed oscillations do indeed result from valley states, as described by the theoretical model in the paper and presented in Fig. 2(b) and Fig. 3. The model appears to be sound, and indeed it shows very good agreement with the experimental data (in Figs. 2(a) & (c)). It clearly appears that there is an excited state in the left dot, and that the energy splitting from the ground state is relatively unaffected by detuning. It is therefore reasonable, at first sight, to interpret this excited state as the upper of the two conduction band valleys.

Perhaps the only outstanding question is whether this excited state might instead be an orbital state rather than a valley state, or some hybrid state of the two. In the Discussion section on page 7 the authors state that:

"... orbital states is precluded by the magnitude of δ . Based on estimates of dot dimension, such an excited state would be expected to be an order of magnitude higher."

Firstly, I think it would be helpful if the authors could provide a simple analysis, perhaps just based on dot size, that would back up this claim. This could perhaps go in the Supplementary Information.

Secondly, I think that perhaps a more compelling argument is that fact that the energy splitting observed for the (1,1)-(2,0) transition is very close to that for the (0,1)-(1,0) transition (both are ~ 6 GHz). If these were orbital excited states then you would expect they would change significantly (most likely decrease) as the electron number increases. The fact the splitting remains constant is very consistent with this being a valley splitting – see, e.g. Jiang et al., PRB 88, 085311 (2013). I suggest that the authors comment on this in their revised manuscript.

I would also add that the paper is clearly presented and quite well written, and I believe that other researchers are well placed to be able to reproduce this work.

So given the significance and novelty of the paper, the fact that the key experimental claims appear to be supported, and that the paper is clearly written, I would suggest that the paper is appropriate for publication in Nature Communications, subject to a few revisions, listed below.

1. As noted above, the authors should comment on the fact that the excited state splitting remains relatively constant for the (1,1)-(2,0) and (0,1)-(1,0) transitions, which provides additional evidence for this indeed being a valley splitting.

2. I think that a simple energy diagram (like that in Fig. 2(b) or Fig 3(b)) should be added to the Supplementary Information in Fig. S1, to show the relevant transitions for the (1,1)-(2,0) case. I don't think it is necessary for the authors to develop a full simulation as they do for the (0,1)-(1,0) case, but it would be helpful to show why the oscillations occur. It would also explain why having a small valley splitting makes it impossible to observe spin blockade at the (1,1)-(2,0) transition here. This also reinforces the significance of the paper.

3. I think the title is a bit cumbersome. I think the words "at Multiple Charge Configurations" could be deleted without losing any significance. If they really think this is an important point for the title, then I would suggest they should move the (1,1)-(2,0) data into the main paper.

4. In Fig. 1(d) the axis is labelled as G, for transconductance (dI_{SET}/dV_g), but the data oscillates about zero, which implies to me that this is actually a differential transconductance, dG/dV_g . Also in the text on page 4 it states that "Fig 1d shows the differential measurement QPC current". This needs to be corrected, or clarified.

5. In Figs. 1(b), 1(c), 2(a) and 2(c), the colour scale is not numerically labelled, so it is not possible to determine what the units are, or the magnitude. This should be fixed.

Response to Referee Reports

Reviewer #1 (Remarks to the Author):

The authors report a very solid and competent set of experiments on a surface-gated silicon structure. They performed good fabrication, and sensitive measurements which they interpret as evidence of the manipulation of valley states in the electrically confined region of the silicon. I think this interpretation is believable, and the data are very clear. They show Rabi oscillations and clear detuning effects. The paper is clearly written. I believe that the results will be of interest to experts in the field, and whilst not providing a viable new qubit technology, due to the small level separation and short coherence time, this work is a useful adjunct to the silicon qubit literature. The issue of valley splitting has been a constant source of discussion in the field, and these results are a step in clarifying that discussion. I recommend that the paper be published as it stands.

We are grateful for the referee's positive comments and support for the publication of the paper in Nature Communications.

Reviewer #2 (Remarks to the Author):

This manuscript presents some interesting data, but has several problems that make it unpublishable in its present form. The biggest problem is that I am not at all convinced that the observed phenomena are attributed correctly to the oscillations of a valley qubit. But there are also deficiencies in the description of the experiment as well as in attribution of previous work that also need to be resolved before the paper could be seriously considered for publication.

We are glad that the referee thinks the data is interesting. He/she however was not convinced that the observed phenomena can be attributed to the oscillations of a valley qubit. We believe that the main confusion was that our description in the text somehow misled the referee to think our qubit operation is carried out in the middle dot. Were this the case, then the observation at multiple charge configurations might not provide strong evidence of valley state manipulation. In the point-by-point response below, we will make clarifications in details. Furthermore, the referee has also identified deficiencies of the paper in the description of the experiments and in citations of previous work, we have now made changes in the text to comply with the recommendations. The corresponding changes will be described in details below.

First I discuss why I doubt that the oscillations are arising from a valley qubit. The stability diagram shown assigns occupations to the dots, but no data shown make the case for these assignments. The key issue here is that if the left and middle dots were both to have one more electron than assigned, then the occupations of the left and middle dot would be (2,1) and (1,2), and the oscillations would be those of a quantum dot hybrid qubit (see, e.g., Refs. 5 and 10 in the manuscript). The stability diagrams are cut off well before further transitions would be observed, but even if they do not yield additional transitions, it is important to show magnetospectroscopy data to provide some confirmation of the occupation assignments.

Indeed, it is important to make sure that the observed oscillations are different from that observed for hybrid charge/spin qubits. We use the fact that oscillations were observed at multiple charge configurations to eliminate the possibility of hybrid qubit.

Experimentally, we are confident that we can identify the last electron in a QD. Even though the charging lines in Fig. 1b are not displayed for an extended range for this particular trace, we have extended to larger range in many cases, for multiple identically fabricated samples. Regarding the magnetospectroscopy, it was not done since the cryostat we used for the experiment doesn't have a magnet. We agree with the referee that without magnetospectroscopy the charge stability diagram cannot be considered a smoking gun showing that there was an absolute number of 1 in a QD for the oscillations. Importantly, the core of our argument does not rely on the absolute occupation number of electrons in the dots. Instead, the evidence in support of our claim relies on having seen left-dot oscillations with a similar operation frequency at two different and consecutive electron occupation numbers within the left dot. If it was confirmed that oscillations occurred at an odd occupation number in the left dot during operation, then we wouldn't even need the second charge configuration to rule out standard spin-charge hybrid qubit operation, as we will discuss further below.

The authors state that an interpretation in terms of a quantum dot hybrid qubit is not tenable because similar phenomena are seen if the occupation of the middle dot is increased by one, but the data shown do not convince me of this. Quantum dot hybrid qubit-type oscillations could still occur after the addition of an extra electron to the left dot (**because the oscillations are occurring between two states in the middle dot, the electron occupation of which is not changing**). It is true that the additional electron on the left could cause a singlet state to form in that dot, but in these systems it is quite possible that the valley splitting for the additional electron is low, in which case it could well be that some spin-related phenomenon such as spin blockade would be observed some of the time and oscillations some of the time, depending on the valley state. This scenario is consistent with the strong signal observed near the polarization line in the presence of the additional electron that is absent when there is no extra electron.

The primary issue with this objection is that the oscillations are actually occurring in the left dot, not in the middle dot, as stated by reviewer #2. And it is the **left dot** whose occupation is increased by one.

We somehow misguided the referee to think we were carrying out the operation in the middle dot. If this were the case, the observation of oscillations at different numbers of electrons in the left dot would not necessary provide evidence that the oscillation is not due to the spin states of the hybrid qubit, where the single dot singlet-triplet splitting provides the bulk of the excited state energy contribution, as the referee pointed out.

To reiterate, the electron begins in the middle dot and then transitions to the left dot, where oscillations occur. This means that the dot in which the oscillations occur and the dot whose occupation number is changing are the **same** dot. If the two QDs were in the hybrid qubit configuration, adding or subtracting one electron from the operating dot would simply disable the oscillations between singlet and triple states.

To avoid confusion, we have now added one paragraph in the beginning of the “principle of valley qubit operation and read-out” section, on page 4, to describe the left dot as the operation QD and the middle QD as the measurement QD.

I don't think that there is a simple explanation of the marked differences in the data near the polarization line in terms of a single-electron valley qubit. This is exactly the argument given by the authors in reverse: if the oscillations are between valley states, then the data at the two fillings should be similar. Since the data are qualitatively different near the polarization line, that is evidence that the oscillations are not between two valley states.

The bright line near the polarization line is due to an incoherent process, which can always be observed with and without the coherent oscillations, for both pulse directions, and for a much large pulse width. When pulsed into the reconfiguration line, the two levels of the QDs are detuned, the electron can then tunnel between the two QDs. Due to the large inter-dot tunneling rate, the probabilities for the electron to be in the two QD are equal. In fact, we found that the strength of this line depends on the inter-dot tunneling rate. Though it is not prominent in the figure from the main body of the paper, which was selected for its relative cleanliness, if requested, we can provide data demonstrating the presence of this feature in the presence of the oscillations at two consecutive anti-crossings. To make a useful note to the readers, we have now added in a paragraph about this bright line in the Supplementary Note 1 and have added a reference on the non-equilibrium pulse effect (Harbusch et al., PRB 82,195310, 2010).

Another reason why I doubt that the oscillations are between valley states is that the time scales of the experiment are much more typical of those of charge and quantum dot hybrid qubits, while the time scales of valley relaxations have been measured to be much longer. As discussed in the manuscript, the relaxation time is probably of the same order as the pulse repetition time, which is reported in the manuscript as about 30 ns; this value is quite similar to those measured in charge and quantum dot hybrid qubits. The decoherence time of about 1 ns is much closer to that observed in charge and hybrid qubits, and much less than the time scales for valley relaxations, which have been measured to be of order of tens and even hundreds of microseconds (see next paragraph).

The direct relaxation used for re-initializing the system is actually charge relaxation, as the ground and excited states no longer share a charge configuration for the measurement point where experiments were carried out, see Fig. 1(a) and Fig S3. In our experiments, the typical charge relaxation is about 10-15 ns. We found 30 ns is long enough for the re-initialization. Another way to re-initialize the system is through tunneling. In fact, we found in certain cases, the tunneling to bath become a fastest relaxation pathway.

The measured dephasing during operation does not reflect on the valley relaxation time, since tunneling effects as well as likely pulse imperfections cause dephasing and oscillation decay at a timescale well below the valley relaxation time, as explained in the supplementary materials. A new paragraph is added in the end of Supplementary Note 2 to stress the point and is refereed on p6. A new figure (i.e., Fig. S3) is along added to show different re-initialization pathways at different working point of the stability diagram.

Now I discuss previous work that is relevant that is not mentioned in the manuscript. The first is that the pulse sequence and measurement interpretation is extremely similar to the "electronic beamsplitter" investigated by Petta et al. (Science 327, 669 (2010), plus follow-on work). I agree that the experiments reported in this manuscript are unlikely to be due to the S-T₋ oscillations explored in the Petta manuscript, but the experimental method as well as the phenomenology of the two cases have a lot of similarities.

We agree with the referee that the paper by Petta et al. is influential to our experimental work. We have added a sentence on page 6 "We would like to point out that the techniques used in this work to probe valley states bear a strong experimental and formal similarity to the electronic beamsplitter methodology used to probe the (1,1)S-(1,1)T₋ of a double quantum dot."

The second set of related work is the work that the Vandersypen lab has done measuring valley relaxations: they have showed that valley relaxation in a single dot is many microseconds (there are two papers, one reporting a valley relaxation time of about 50 microseconds (E. Kawakami et al., Nat. Nano. 9, 666 (2014)) and the other reporting a valley relaxation time of hundreds of microseconds (PNAS 113, 11738 (2016))).

The two references are now cited in a newly added paragraph on p5 to refer to the very long valley relaxation times reported in the literature.

To summarize so far, because I think that the interpretation of the experiments is wrong, I do not recommend publication of the current version of the manuscript in any journal.

We hope our clarification of the confusion and our explanations will convince the referee that our interpretation is sound. We are confident in our explanation. In fact, we have done experiments with a string of different devices. Valley splitting varied from 0.3MHz to 7GHz have been realized in different devices, again for multiple charge configurations. These results will be reported in a future paper on Ramsey fringe measurements, with additional collaborators.

I now move on to general issues that should be addressed to make the set of measurements more complete and interpretable.

1) Magnetospectroscopy measurements should be shown, to provide better evidence of the electron filling factors at which the experiments are performed.

As we have mentioned before, magnetospectroscopy is not feasible with our dilution fridge setup and our main experimental findings do not rely on the absolute filling factors at which the experiments were performed, but rather on the fact that they were observed at consecutive filling factors, with the oscillation happening within the operation dot whose occupation number was changing.

2) The relaxation time relevant for readout should be measured, to be able to give some sense of the magnitude of the oscillations compared to one electron moving from one side of the double dot to the other. The authors state that the relaxation rate may be determined by tunneling out of one dot and into the other. If they have experimental evidence for this claim, it should be shown.

If the process is a relaxation process within a single dot, the dependence of the relaxation rate on detuning will yield insight into the underlying physics (see, e.g., Wang et al., Phys. Rev. Lett. 111, 046801 (2013); Nguyen et al., arXiv:1403.3704). In either case, the dependence of this relaxation rate on detuning should be characterized in order to obtain some information about the strength of the oscillations as the detuning is changed.

Based on the suggestion of Reviewer 2 we have included an analysis of the relaxation pathways during the measurement period as a function of the measurement point (Figure S3 and discussion). As it was suggested it would, this analysis adds insights regarding the mechanisms of relaxation as well as the valley to valley relaxation time.

As we have explained earlier, because of this long relaxation time, the observed coherent oscillations in the operating phase is not affected by the valley relaxation time. Instead, it is the short charge relaxation or tunneling times that determines our signal decay in the measurement phase. Therefore, the measurement of valley relaxation time of the operation dot cannot be done using the technique described in the paper. The evidence of re-initialization by tunneling is discussed in the newly added paragraph in Supplementary Note 2 and Fig. S3.

3) The main text should state that the signal being plotted is the real part of the output of the lockin amplifier (at present this is stated in the supplementary material, but not the main text). The frequency of the modulation should be reported.

Following this advice, we have moved this info to the main text. The frequency of modulation is now reported.

A confusing typo:

Supplemental Fig. S1:

(2,0,1) appears twice; I assume that the lower one should be (2,0,0) (modulo my doubts about the assignments made alluded to above).

Thank the referee for spotting the typo. It is now fixed.

In conclusion, we thank the referee for providing many insightful and detailed remarks, which helped us to improve the quality of our manuscript. We hope that the referee finds the revised manuscript clarifies confusions and addresses his/her concerns and recommends for its publication.

Reviewer #3 (Remarks to the Author):

This paper claims to observe the coherent manipulation of valley states in a silicon quantum dot by the application of fast gate control pulses to a triple quantum dot system, although in this case only two of the quantum dots appear to contribute to the effect, since the third dot is maintained at a constant occupancy throughout the experiments. The experiment also claims to provide a method to extract the valley splitting with high accuracy.

I believe that there is sufficient evidence in the experimental results to support these claims, but I will return to this point later, and firstly consider the overall potential significance of the paper.

Silicon-based quantum dots are generating significant interest as qubits for future quantum computing systems because of the long spin coherence times that are available in silicon and their compatibility with industrial silicon CMOS manufacture. In many cases, such spin qubits rely on the process of Pauli spin blockade to discriminate between singlet and triplet two-electron states, which can either be the qubit basis states, or serve as a pathway for readout of single spins.

As the authors correctly note, the Pauli spin blockade process requires a sufficiently large valley splitting, but many silicon quantum dots, in particular those formed at Si/SiGe heterostructures, can have very small valley splittings (much less than 0.1 meV) which make singlet-triplet readout impossible. While this is not a fundamental obstacle to silicon spin based QC (since a number of groups have now demonstrated spin blockade), it is proving to be an issue that is slowing progress in the field, and is therefore of considerable practical interest to the silicon QC community.

I therefore believe that the spin qubit community will find this paper both interesting and useful, and the experiments and analysis here will provide a useful tool to characterise small valley splittings when they occur.

I also believe that this work is indeed novel, as I am not aware of any other experiments which have shown coherent oscillations that can be attributed to valley states.

We are very grateful to referee's strong supportive statements. We also appreciate the referee for pointing out the significance of the work.

I now return to the question of whether the observed oscillations do indeed result from valley states, as described by the theoretical model in the paper and presented in Fig. 2(b) and Fig. 3. The model appears to be sound, and indeed it shows very good agreement with the experimental data (in Figs. 2(a) & (c)). It clearly appears that there is an excited state in the left dot, and that the energy splitting from the ground state is relatively unaffected by detuning. It is therefore reasonable, at first sight, to interpret this excited state as the upper of the two conduction band valleys.

Perhaps the only outstanding question is whether this excited state might instead be an orbital state rather than a valley state, or some hybrid state of the two. In the Discussion section on page 7 the authors state that: "... orbital states is precluded by the magnitude of δ . Based on estimates of dot dimension, such an excited state would be expected to be an order of magnitude higher." Firstly, I think it would be helpful if the authors could provide a simple analysis, perhaps just based on dot size, that would back up this claim. This could perhaps go in the Supplementary Information.

Following referee's advice, we have added a paragraph to estimate the size of the orbital excited state. In short, for a charging energy of 2.5meV, a dot radius of 40 nm is expected which would correspond to the first excited orbital state of 0.4meV above the ground state (i.e., 93GHz much larger than the 5-6 GHz qubit oscillation frequency).

Secondly, I think that perhaps a more compelling argument is that fact that the energy splitting observed for the (1,1)-(2,0) transition is very close to that for the (0,1)-(1,0) transition (both are ~ 6 GHz). If these were orbital excited states then you would expect they would change significantly

(most likely decrease) as the electron number increases. The fact the splitting remains constant is very consistent with this being a valley splitting – see, e.g. Jiang et al., PRB 88, 085311 (2013). I suggest that the authors comment on this in their revised manuscript.

This is indeed a very good argument supporting the energy splitting is not due to orbital excited states. We have added sentences in p7 to provide this argument and have added a reference to the report that valley-orbit coupling is unaffected by the occupation number.

I would also add that the paper is clearly presented and quite well written, and I believe that other researchers are well placed to be able to reproduce this work.

So given the significance and novelty of the paper, the fact that the key experimental claims appear to be supported, and that the paper is clearly written, I would suggest that the paper is appropriate for publication in Nature Communications, subject to a few revisions, listed below.

We really appreciate the recommendation!

As noted above, the authors should comment on the fact that the excited state splitting remains relatively constant for the (1,1)-(2,0) and (0,1)-(1,0) transitions, which provides additional evidence for this indeed being a valley splitting.

This is a very good point. We have added one paragraph in page 7 to provide this additional evidence.

2. I think that a simple energy diagram (like that in Fig. 2(b) or Fig 3(b)) should be added to the Supplementary Information in Fig. S1, to show the relevant transitions for the (1,1)-(2,0) case. I don't think it is necessary for the authors to develop a full simulation as they do for the (0,1)-(1,0) case, but it would be helpful to show why the oscillations occur. It would also explain why having a small valley splitting makes it impossible to observe spin blockade at the (1,1)-(2,0) transition here. This also reinforces the significance of the paper.

Following the suggestion, we have added an energy diagram as an additional figure (Fig. S1e) for the (1,1)-(2,0) case. Since the operation is carried out in the (2,0) state and measured in the (1,1), as oppose to the spin-qubit case, the lift of spin blockade unfortunately cannot be discussed using this diagram.

3. I think the title is a bit cumbersome. I think the words “at Multiple Charge Configurations” could be deleted without losing any significance. If they really think this is an important point for the title, then I would suggest they should move the (1,1)-(2,0) data into the main paper.

We would like to keep these words in the title. The observation at multiple charge configurations is an important piece of proof for our claim, as we described in the response to referee 2.

4. In Fig. 1(d) the axis is labelled as G, for transconductance (dI_{SET}/d_Vg), but the data oscillates about zero, which implies tome that this is actually a differential transconductance, dG/d_Vg . Also

in the text on page 4 it states that “Fig 1d shows the differential measurement QPC current”. This needs to be corrected, or clarified.

The transconductance signal is centered around 0 because it was background subtracted using a moving average, not because it was differential. This is mentioned in the caption.

5. In Figs. 1(b), 1(c), 2(a) and 2(c), the colour scale is not numerically labelled, so it is not possible to determine what the units are, or the magnitude. This should be fixed.

Thank the referee for spotting the oversight. We have now added the color scale into the figures.

In conclusions, we are very grateful for the strong support the referee has given us to publish the paper. The changes and additional information based the recommendations indeed have added additional support for our claim that the valley states are the cause of the coherent oscillations.

REVIEWERS' COMMENTS:

Reviewer #2 (Remarks to the Author):

The authors have done a very good job of addressing the concerns that I raised in my first report. I am now convinced that they have strong evidence that they have a valley qubit, and that it is interesting. I am happy to recommend publication.

Reviewer #3 (Remarks to the Author):

I have looked over the point by point response letter to the referee reports, and the revised manuscript, and I believe that the points raised in the previous round of review have been satisfactorily addressed.

I therefore recommend publication of the revised manuscript in Nature Communications.

Response to Reviewer's Comments

REVIEWERS' COMMENTS:

Reviewer #2 (Remarks to the Author):

The authors have done a very good job of addressing the concerns that I raised in my first report. I am now convinced that they have strong evidence that they have a valley qubit, and that it is interesting. I am happy to recommend publication.

Reviewer #3 (Remarks to the Author):

I have looked over the point by point response letter to the referee reports, and the revised manuscript, and I believe that the points raised in the previous round of review have been satisfactorily addressed.

I therefore recommend publication of the revised manuscript in Nature Communications.

We are delighted that the referees have made positive recommendations to publish the revised manuscript.